# Digital Trends, Digital Literacy, and E-Health Engagement Predictors of Breast and Colorectal Cancer Survivors: A Population-Based Cross-Sectional Survey

**DOI:** 10.3390/ijerph20021472

**Published:** 2023-01-13

**Authors:** Samar J. Melhem, Shereen Nabhani-Gebara, Reem Kayyali

**Affiliations:** Department of Pharmacy, School of Life Sciences, Pharmacy and Chemistry, Kingston University London, Kingston upon Thames, Surrey KT1 1LQ, UK

**Keywords:** cancer, survivor, digital literacy, e-health literacy, digital divide, Internet, information, Arab, Jordan

## Abstract

Introduction: Advances in information and communication technology (ICT) and post-COVID-19 tectonic changes in healthcare delivery have made it possible for cancer survivors to obtain disease-related information for remote management online rather than through healthcare providers. To comprehend and evaluate health information, digital literacy is crucial. Objectives: This study examined cancer survivors’ information-seeking behaviour, information sources, digital health literacy, and digital trends, as well as potential determinants of e-health information receptivity and online resource use. Methods: A national 30-item cross-sectional survey using a representative random sample of cancer survivors from Jordan’s cancer registry was conducted. Chi-square tests established categorical variable relationships. Using the mean and standard deviation, we calculated the Likert scale’s ordinal data average. A *p*-value < 0.05 was statistically significant. Logistic regression identified predictors of interest in late-trajectory information acquisition and use of e-health platforms (apps, portals) for cancer self-management. Results: Lower digital literacy and electronic searching were associated with older age and lower income, education, and employment status (*p* ≤ 0.001). Digital literacy independently predicted m-health app use for remote management and interest in cancer supportive care information. Digitally literate survivors preferred the use of digital platforms (*p* ≤ 0.001). Information acquisition barriers included “reliability” (26%, *n* = 25) and “health information trustworthiness” (16.2%, *n* = 25). Following treatment completion, Internet-seeking behaviour decreased significantly when compared to the early cancer trajectory. Conclusion: Our findings imply that Jordanian cancer survivors’ low digital literacy may hinder information acquisition and technology-enabled cancer care. Digital interventions for cancer survivors should be adaptable to varying levels of digital health literacy. Healthcare policymakers should recognise digital inequities and devise focused initiatives to bridge the digital divide while responding to the urgent need to digitalise cancer care delivery.

## 1. Introduction

Cancer is one of the most-searched medical conditions on the Internet due to its global prevalence [1]. Among cancer survivors, Internet information acquisition increased from 53.5% to 69.2% in 2017 [2]. Cancer survivors have a wide range of symptoms, including psychosocial, cognitive, physiological, and sexual issues, as well as concerns about recurrence and secondary neoplasms. Cancer patients have more complicated health information requirements than healthy subjects, and insufficient information or misinformation may lead to information dissatisfaction or suboptimal outcomes [3]. Patients may seek alternative sources of information if information provided by healthcare professionals is inadequate [4,5]. Even if patients are satisfied with the care or information they have received, they may seek additional information from other sources. Arabic-speaking cultures have little data on breast and colorectal cancer patients’ digital trends, preferred sources of information, and online information seeking [6]. Alnaim in 2019 [7] examined the quality of breast cancer information on Arabic websites and found that while many exist, few are moderated by experts, the majority provide poor quality information, and only two-thirds provide completely correct information. Online cancer information helps survivors learn about treatment options, fulfil their cancer-related information needs, self-educate themselves about the disease, and get practical advice from peer patients [4,5]. Using the Arabic Internet for cancer-related information can be problematic because much of it is unaccredited or unvalidated by specialists [7,8]. Health information is increasingly being searched for on mobile devices and social media [9] Misinformation resulting from patients’ increased uploading of cancer data to social media, blogs, and networks raises doubts about the accuracy of online information [10,11,12,13]. The COVID-19 pandemic has accelerated the flow of chronic illness information online, including cancer. According to the World Health Organisation (WHO), this has led to an infodemic (information pandemic) characterised by an abundance of online and offline information. The infodemic has resulted in a lot of true, false, and mixed information on the Internet. Non-evidence-based, untrustworthy, erroneous, deceptive, or misleading data might make information acquisition more challenging [7,12,13,14,15]. The vast amount of cancer information available online can be overwhelming, confusing, and disturbing for some patients, especially if they are unable to filter and digest it [16,17,18]. Further, the COVID-19 pandemic has forced changes in health services delivery, accelerated digital solutions with rapid shifts toward remote patient care delivery, and increased online health-information seeking. Technology’s growing role in healthcare delivery has revealed health systems’ fragility and global digital divides. The implications of post-pandemic technological shifts on vulnerable patient populations in low- and middle-income countries is a major challenge [19]. In this setting, using information and communication technology and digital health to improve direct doctor–patient communications and implementing creative service delivery models will be crucial to overcoming these challenges [20]. However, for wide-scale successful adoption and implementation of these innovations, the emphasis on a patient-centred approach is a critical success factor. This should be considered to mitigate potential flaws that could worsen the fragility of developing countries’ health care systems and exclude vulnerable and marginalised populations [19]. Although the mobile communication sector has grown significantly in low-and-middle income countries (LMICs) over the past decade, women, rural communities, and disadvantaged populations still face disparities in access to digital services, cell phones, the Internet, and media coverage [21].

In recent decades, the digital divide was defined by technology access disparities based on age, education, socioeconomic status, and ethnicity [22,23]. Cancer survivors with lower digital literacy had less access and gained less from searching online resources than those with higher digital literacy [24]. Low digital literacy also negatively impacts health information interpretation and trust in online health resources [25]. The Internet’s potential to make health care information accessible is tempered by its drawbacks. In the midst of conflicting health information on the Internet, finding reliable sources can be difficult [26].

Alternatively, digital health is “a broad scope which covers mobile health (mHealth), health information technology, wearable devices, telehealth and telemedicine, and customised treatment,” as defined by the US Food and Drug Administration [27]. As digital health becomes more popular, digital literacy and health literacy are increasingly important elements in evaluating its efficacy. Digital literacy is “the capacity to use information and communication technology to access, analyse, generate, and exchange information, which needs both cognitive and technical abilities” [28].

“Digital health literacy” evolved from “eHealth literacy” over time, the term “eHealth literacy” is included under the umbrella of “digital health literacy.” Norman [29]. defined eHealth literacy in 2006 as the ability to seek, obtain, interpret, and evaluate health information via electronic channels and apply the knowledge learned to solve health problems. Thus, digital health literacy influences health status, health disparities, the digital divide, and public attitudes and practices. The understanding of digital health may be addressed by improving population-level digital health literacy [28]. Additionally, increasing digital health literacy can help address new health concerns. The COVID-19 pandemic has highlighted the importance of digital health literacy in finding cancer-related information online [30]. In this context, the ability to obtain and analyse information using new technologies’ is crucial. For example, advancements in information and communications technology (ICT) have made it feasible for patients to easily acquire disease-related or health-related information online rather than from healthcare providers during outpatient consultations [9]. Following this trend, the ability to appropriately grasp and assess health-related information on the Internet is vital. Therefore, understanding the current state and trends in digital health literacy research and identifying future research opportunities is crucial. There is scant evidence about Arab and Jordanian cancer survivors’ digital trends and online information-seeking behaviour. To fill this gap, this study aimed to (a) explore cancer survivors’ information-seeking behaviour, information sources, digital health literacy, and digital trends, and (b) identify possible predictors of e-health information receptivity and use of online resources.

## 2. Methods

The present study examines Jordanian breast and colorectal cancer patients’ online information-seeking behaviour, digital trends, and barriers to information acquisition as part of a larger project by Melhem et al. in 2022 [3]

### 2.1. Study Design and Setting

A population-based cross-sectional survey was conducted between 1 March and 17 July 2020. The sample population was derived from all alive Jordanian breast and colorectal cancer survivors diagnosed in 2015–2016 who reside in Jordan and meet all the predefined eligibility criteria.

### 2.2. Study Population

The study’s target population consisted of all Jordanian breast and colorectal cancer patients of both sexes registered by Jordan’s national cancer registry (JCR) in 2015 and 2016 and residing in Jordan. We opted not to include patients without a national identification number (ID) to rule out the possibility of a systematic bias in the data brought on by non-Jordanian cancer patients living in Jordan who are registered in the national cancer registry. The following procedure was carried out to define the study population:

Step 1: Defining the population frame

In 2015 and 2016, the national cancer registry registered a total of 3736 breast and colorectal cancer patients from all nationalities (Jordanians and non-Jordanians) which constitute the crude study population frame.

Step 2: Refining the population frame

After consulting the department of morbidity and mortality of the noncommunicable diseases directorate (NCD) at the Ministry of Health (MOH) and excluding patients who were not Jordanians or who had gone away, the total number of patients as of 30 December 2019 was 2487.

Step 3: updating the population frame

The study population was updated due to inadequate or outdated contact information. This was done in collaboration with Jordan’s civil department between 1 January and 31 March 2020. As a result of completing this phase, there were 1567 patients with verified phone numbers who remained alive until 29 February 2020. This was considered the eligible population since they could be reached in an efficient and practical manner.

### 2.3. Participants and Recruitment

The study population was derived from Jordan Cancer Registry’s (JCR) database in 2015–2016. The study population comprised 1567 adult survivors (≥18 years). Inclusion criteria include all the following: being a Jordanian citizen, alive until 29 February 2020, and having correct contact details. Non-Jordanian citizens, those who were living abroad at the time of data collection, or those who were unreachable because of missing contact information were excluded.

### 2.4. Sampling Procedure and Randomisation

The Krejcie and Morgan equation determined a representative statistical sample size of 309 individuals [31]. The planned sample size was augmented by 30% (*n* = 409) to account for anticipated non-response (due to death, rejection, error in phone number, etc.). Response proportions corresponds to the responded study sample/eligible sample size of the population frame (n/N). Proportional distribution of size was used to distribute the sample to males and females for each type of cancer; the sample size distribution per cancer type and gender This sample constituted a representation of the three cancer types included based on their prevalence. Sample characteristics is shown in Table 1.

The sampling method used in this study is the probability sampling method which is random sampling. Probability sampling is defined as one in which every unit in the population has a chance (0 < x < 1) of being selected in the sample, which can be accurately determined. Using SPSS (Statistical Package for the Social Sciences) version 22, a representative systematic random sample was generated from the entire ranked population frame (1567) by age, gender, and type of cancer in ascending order. The first subject was chosen randomly from a table of random numbers, and the remaining subjects were chosen automatically using an explicit sampling frame according to a predetermined sampling interval (k = 4).

Krejcie and Morgan equation is given by the following formula:

S = *X*^2^*NP* (1 − *P*) ÷ *d^2^* (*N* − 1) + *X*^2^*P* (1 − *P*)

S = required sample size.

*X* = the table value of chi-square for 1 degree of freedom at the desired confidence level (1.96)

*N* = the population size.

*P* = the population proportion (assumed to be 0.50 since this would provide the maximum sample size).

*d* = the degree of accuracy expressed as a proportion (0.05).

### 2.5. Survey Design

To inform the questionnaire design, a literature review was conducted to identify factors likely to influence online information-seeking behaviour and digital literacy [32,33,34,35]. The questionnaire (Appendix A) had four sections and 30 items in the domains of sociodemographic information, online information access and search, cancer supportive care information, and survivorship information openness. Cancer survivorship refers to a patient’s health and well-being from the time of diagnosis until death. This covers the physical, mental, emotional, social, and economic consequences of cancer, which begin with diagnosis and continue through treatment and beyond [3,36]. Section one included eight tick box items that assessed respondents’ socio-demographic characteristics, including age, gender, residence, marital status, employment, monthly income, education, and comorbid illnesses. The second section of the questionnaire focused on evaluating the use of the Internet to access and search for cancer supportive care information. The section had five questions. The first asked respondents to rate their digital or e-health literacy defined as “*the ability to search, find, understand, and evaluate health information from electronic or online sources and apply the knowledge to solve health-related problems*,” on a five-point Likert scale from 5 (very good) to 1 (very poor). Smartphone ownership was the second yes/no question. If yes, the participant was asked which mobile apps they use in a multiple-choice question. Finally, a tick box question was added to see if patients are open to receiving cancer-related information for cancer management and doctor–patient communication via a mobile app. Section three examined breast and colorectal cancer information from online sources with 11 items. This section asked patients if they personally used the Internet to find cancer-related information or if anyone around them did since their diagnosis. If they answered “yes” to the latter question, they were asked to list who in their circle used the Internet for information. Next, a multiple-choice question asked which health information sources they used, and which was most useful, leaving a blank space to answer. If the Internet was identified as a source of information, the survey asked the participant which online sources they used in a multiple-choice question, and which was most useful in a blank space. After that, patients who used the Internet were asked about the frequency of information seeking from diagnosis to treatment completion compared to survivorship. The questionnaire then asked participants about their reasons for searching for online cancer-related information sources, obstacles they faced, and how they critically assessed the reliability/trustworthiness of the information they found using multiple choice answers. Respondents who had not used the Internet for cancer information were asked to explain why using a multiple-choice question. The questionnaire ended with a yes/no question to determine if breast and colorectal cancer patients are still interested in receiving information to help them manage their condition. If the participant answered “yes,” a multiple-choice question asked about their preferred method of receiving such information. Finally, a (yes/no) question assessed patients’ willingness to use a mobile app for information needs. If they answered “no,” they could write their reasons.

### 2.6. Piloting Survey Questionnaire

The questionnaire was piloted at Jordan University Hospital (JUH), a semi-government tertiary hospital in Amman, from January to March 2020. The pilot included 26 breast and colorectal cancer patients (22 females and 4 males) of various ages, educational levels, and socioeconomic backgrounds. The pilot study assessed the questionnaire’s length, language clarity, comprehensibility, and format, as well as participant feedback for further refinement [37,38,39]. Since the survey was originally designed in English, linguistic and cultural validation were conducted using the forward/backward translation technique to accommodate the target population’s language proficiency [40]

### 2.7. Analysis

Data was coded and entered into pre-designed Microsoft Excel sheets. Data consistency and completeness were verified. The analysis used IBM SPSS version 22. To present categorical data, descriptive statistics were used (i.e., percentages, frequency, mean and standard deviation). Chi-square test was used to establish categorical variable associations. The overall mean of the Likert scale-generated ordinal data was calculated using the mean and standard deviation. Logistic regression was used to identify the predictors of online information seeking during cancer survivorship and involvement in e-health platforms (apps or portals) to receive self-management cancer information. A *p* value < 0.05 was set for statistical significance.

### 2.8. Ethical Approvals

The study was approved by Jordan’s Ministry of Health (MBA/ethics committee. /21115), University of Jordan (10/2019/8990), and Kingston University’s ethical requirements for scientific research (Approval number\2885). A participant information sheet (PIS) outlined the study’s goals for each participant. The phone interview implied consent. In accordance with JCR regulations, participants’ confidential and anonymous information was secured, kept private, and used only to achieve the study’s goals.

## 3. Results

### 3.1. Participant Characteristics

In total, 409 patients were asked to participate, and 335 answered (81.9% response rate). The population characteristics are summarised by Melhem et al [3]. and below. Nearly a third, 31% (*n* = 105) were 60 or older, and 35% (*n* = 116) were 50–59. The sample’s median age was 55 years old (males 62.5 years, ladies 55 years). Female participants outnumbered male participants by 84% (*n* = 281) to 16% (*n* = 54). Breast, colon, and rectal cancer accounted for 76.1%, 17.6%, and 6.3%, respectively. Rectal cancer patients had the highest response rate 80.8%, followed by breast cancer 76.1% and colon cancer 73.8%. Non-responders included 21 patients (5.1%) who declined to engage in the survey, 5 patients (0.01%) who died during data collection, 11 patients (1.45%) whose phone numbers were inaccurate or disconnected, and 2 patients (0.48%) who were out of the country during survey conduction. Housewives comprised 72.5% of the female respondents with 20.3% of respondents being retired and 17.1% employed. Of the majority, 61.2% lived in Amman, the capital, and 82.1% were married. When asked about their monthly household income, 38.2% of patients either declined to answer or indicated they “don’t know.” Of those who responded, 43.3% stated it was less than 700 dollars (500 Jordanian Dinars (JDs) each month), while 18.5% said it was more than ((700$) 500 JDs). Nearly half of those surveyed 47.1% have other chronic illnesses, including 33.2% having hypertension, 22.7% with diabetes, 8.8% having cardiovascular disease, and 7.6% having other chronic diseases. Nearly one fifth, 21.8% of patients, had a Master’s/PhD/university degree, 27.2% had completed high school, 20% had a diploma, 23.3% had elementary education, and 7.8% were illiterate [3].

### 3.2. Mobile App Ownership and Apps Use

The findings indicate that 94.9% of cancer survivors own smartphones (n=318). Mobile apps were used by 74.9% (*n* = 250). Health apps were used by 14.6% (*n* = 37), e-services by 38.9% (*n* = 98), and social media by 99.6% (*n* = 251).

### 3.3. Internet Use for Cancer-Related Information and Influential Factors

Almost half of the participants (45.9%) looked up cancer-related information on the Internet. Nearly two-thirds (61.5 %, *n* = 206) of individuals asked stated that someone from their immediate circle had searched for cancer-related material since their diagnosis. Of those, 99.5% (*n* = 205) said their family members had checked the Internet for cancer information, compared to 2.9% who said their friends had. Bivariate analysis (Table 2) showed that cancer survivors who accessed online information were younger, more educated, had a trend toward greater income, and had a comorbid condition (e.g., diabetes, hypertension, cardiovascular disease). There was a significant difference in online information use between cancer types (*p* = 0.02), with 82.1% (128) of breast cancer patients actively seeking online cancer-related material, 13% (20) of colon cancer patients using the online sources to gain information (perhaps due to the older demographics of these populations), and only 6 (3.9%) of rectal cancer patients visiting online cancer websites. Higher-income cancer patients were more likely to use online resources (*p* < 0.001). In addition, employed patients were more likely to use them than unemployed patients (*p* < 0.001). The higher-educated patients (masters/PhD, university degree) were more inclined to search for online resources than those with a lower education (*p* < 0.001). However, gender and area of residence had no effect on the likelihood of using online cancer resources (*p* ≥ 0.05 and *p* = 0.1, respectively).

Employment status, marital status, educational level, and cancer type all influenced smartphone ownership (*p* < 0.001) but not gender (*p* = 0.06), age group (0.79), residence (0.60), or monthly income (0.22). However, except for location of residency (*p* = 0.430), Chi square tests showed that mobile app users were considerably more educated (*p* < 0.001), more likely to be female (*p* = 0.003) and diagnosed with breast cancer (*p* < 0.001). They were more likely to be younger (*p* = 0.001), to have a greater monthly income (*p* = 0.001), and to be employed (*p* = 0.001). No significant association was found between gender (*p* = 0.15) and residence (*p* = 0.21) and participants’ desire to receive cancer information via a mobile app. Non-comorbid patient were more likely to use a mobile app to research breast and colorectal cancer than those with chronic diseases (e.g., diabetes, hypertension, and cardiovascular disease) (*p* < 0.001).

### 3.4. Online Cancer Information Acquisition across Cancer Continuum

Patients who used the Internet for cancer research (*n* = 156) were examined for Internet-use frequency across the cancer trajectory. Nearly 56.7% (*n* = 88) of respondents used the Internet to find cancer information daily or 1–3 times per week during the early stages of the cancer continuum, from diagnosis to treatment. However, 63.9% (*n* = 99) of patients reported using the Internet less than once a month or once a year after treatment completion (1–3 times a year).

### 3.5. Barriers to Accessing Online Cancer-Related Information

“Reliability” (26%, *n* = 40) and “health information trustworthiness” (16.2%, *n* = 25) were the biggest obstacles patients faced when searching for information online. Additionally, 10.4% (*n* = 16) of the respondents said that “information was available in other languages that they don’t speak or understand,” while 15.6% (*n* = 24) of the respondents said that “the information was not tailored to their individual needs.” However, 35.7% (*n* = 55) of patients reported no difficulties in finding information. Interestingly, most patients were critical of the material they searched for online and took steps to verify its authenticity, with 63% (*n* = 97) indicating that they verified the accuracy of the information with their doctor, 25.3% (*n* = 39) verifying the results on other websites, and 18.2% (*n* = 28) passively accepting the online search results without further verification (Table 3).

### 3.6. Association between Digital Literacy and the Demographics Breast and Colorectal Cancer Survivors

Of 335 respondents, 39.4% (*n* = 132) rated their digital literacy as “poor” or “very poor”, 46.9% (*n* = 157) as “good” or “very good”, and 13.7% (*n* = 46) as “acceptable”.

Age influenced respondents’ digital literacy (Table 4). Older patients over 60 were 73% more likely to be digitally illiterate than younger patients under 49 (*p* < 0.0001). Digital literacy also increased with employment status (*p* < 0.001). Higher monthly income was associated with higher digital literacy (*p* < 0.0001). The respondents’ digital literacy was also significantly influenced by educational attainment (*p* < 0.0001). Self-rated digital literacy was unrelated to gender (*p* = 0.884), cancer type (*p* = 0.546), or residence (*p* =0.58). In general, demographic factors that increased digital literacy also increased cancer survivors’ online information seeking.

### 3.7. Digital Trends of Breast and Colorectal Cancer Survivors

Digital literacy influenced respondents’ use of online breast and colorectal cancer resources (*p* < 0.001) (Table 5). Compared to 73.4% (113/154) of patients with “good” and “very good” digital literacy skills, more than half (63%, 114/181) of those with “poor” and “very poor” digital literacy had not searched for online resources. In addition, cancer survivors with higher self-reported digital literacy scores (“good” and “very good”) were more interested in receiving cancer-related information during survivorship (*p* < 0.001). Only 27.5% (63/229) of those with the lowest digital literacy scores (“poor” and “very poor”) expressed interest in receiving cancer-related information during survivorship, compared to 56.4% (128/229) of those with higher scores. Higher digital literacy survivors (61.0 % (140/229)) were more likely to use an educational cancer app for remote management during survivorship than lower digital literacy survivors (*p* < 0.001). Table 5 and Figure 1 exhibit survivors’ digital literacy trends.

### 3.8. Further Interest in Information

The study also examined the percentage of patients who are still interested in obtaining information to assist them in controlling their disease at the time of the survey; 68.4% of the patients expressed an interest in doing so and expressed a willingness to use a mobile app if it could meet their information needs. Less than half, 48.1% of patients preferred obtaining cancer-related information verbally from their doctors, 41.2% via mobile apps, 38.6% online, and 7.7% via booklets or brochures. However, 31.6% said they no longer want more information or want to use a mobile app. Digital information inertia during survivorship is attributed to “being digitally illiterate”, “doesn’t know/like using the Internet to gain information”, “no longer interested in getting information about cancer”, “do not want to be reminded of the disease”, “feeling depressed & doesn’t need more information”, “prefer face-to-face communication with attending doctor”, “Internet-based information could make them anxious”, and “Finished therapy and cured, no need for information”.

Unwillingness to use a mobile app to receive cancer-related information for remote management was cited for the following reasons “doesn’t know/like using apps to get information”, “mobile app may be a constant reminder of the disease”, “unsure of usefulness, trustworthiness or unsure of being able to need/use the app”, “don’t want to actively search for information at this stage but still can use app to receive personalised information”.

### 3.9. Predictors of M-Health Engagement and Online Information-Seeking during Survivorship

Table 6 shows the logistic regression model of the predictors of online information interest among breast and colorectal cancer survivors across the cancer continuum. Based on the Omnibus Tests of Model Coefficients”, it was identified that the overall model has a significant impact on the prediction of cancer patients’ receptiveness of receiving cancer-related information to help them in self-managing their condition in survivorship (Chi-square test (df = 22) = 88.031, *p* < 0.001). According to the Nagelkerke R square, the independent variables in the current model can explain 46.2% of the total variation of the dependent variable. Patients with the lowest self-reported digital literacy (i.e., very poor) were 80.6% (AOR (95% CI) = 0.194 (0.060, 0.623)) less interested in receiving supportive care information for self-management relative to those with very good digital literacy (*p* value: 0.006).

Digital literacy was found to be the only independent predictor of cancer-related information acquisition during survivorship (*p* value < 0.001).

Table 7 shows the logistic regression analysis of the predictors of m-health app uptake by breast and colorectal cancer survivors for self-management. The overall model is predictive of m-health app uptake for self-management by breast and colorectal cancer survivors, according to the “Omnibus Tests of Model Coefficients” based on Chi-square test (df 22 = 116.632, *p* < 0.001). The current model’s independent variables explain 57.4% of the dependent variable’s variation, according to the Nagelkerke R square. Although age was not an independent predictor of m-health app uptake (*p* = 0.218), the regression model shows that survivors with ages (50–59) years (*p* = 0.045) and (60–69) years (*p* = 0.043) were 3.4-times more likely to adopt m-health apps for self-management; AOR (95% CI) = 3.386 (1.030, 11.130) and AOR (95% CI) =3.353 (1.039, 10.824), respectively than those over 70 years. Regional residence independently predicted m-health uptake (*p* = 0.029), with survivors in urbanised regions (the middle of the country) being thirteen times; AOR (95% CI) = 13.285 (1.793, 98.414) more likely to use an app for informational support than those in rural areas (southern region). Digital literacy also acted as an independent predictor of m-health app usage (*p* = 0.000). Cancer survivors with very low digital literacy were 98.7% less likely to use an app for cancer self-management during survivorship than those with the highest self-reported digital literacy (AOR (95% CI) = 0.013 (0.002, 0.112)). Similarly, survivors with self-reported poor, acceptable, and good digital literacy were 94.8% (AOR (95% CI), 0.052 (0.004, 0.635)), 91.1% (AOR (95% CI) = 0.089 (0.101, 0.816)), and 91.8% (AOR (95% CI) = 0.082 (0.010, 0.658)) less likely to use m-health tools for cancer supportive care than those with the highest self-reported digital literacy.

## 4. Discussion

This study examined Jordanian breast and colorectal cancer survivors’ online information seeking and digital trends. According to a systematic review [41], mobile health apps empower disadvantaged patients and improve their health literacy by enabling them to communicate with their medical teams. Digital literacy includes general health literacy and online information acquisition usability/navigation skills [42]. Language, culture, and health literacy skills should be considered when designing and developing mobile health apps, especially for low-income and ethnic minority groups [43]. Despite the growing interest in developing mobile health apps for cancer supportive care, patient education, and self-management [44]. There is little emphasis on digital literacy to ensure their optimal use. As a result of the ongoing digital transformation of health care systems, digital health literacy is becoming increasingly significant, to the point where it can be regarded a prerequisite for actively engaging in the systems of the present and future. In the first framework of the World Health Organization’s (WHO) to depict the “Digital Health Ecosystem”; digital health literacy was included among digital health, telemedicine, big data, and governance activities. In the previous five years, the WHO has produced several important papers, including Digital Technologies: Shaping the Future of Primary Health Care (2018) and the first WHO guideline recommendations on Digital Interventions for Health System Strengthening (2019) [45]. Digital health will surpass the internet as a means of providing cancer patients with digital informational support in the future [46].

The findings showed that digital literacy was linked to socio-demographic factors like age, education level, monthly income, and employment, as well as online cancer information searches. Lower digital literacy was linked to older age, lower monthly income, lower education, and unemployment. Digital literacy was unrelated to cancer type. The findings showed that higher digital literacy was linked to a higher likelihood of using online resources to search for cancer-related information since diagnosis and may increase receptivity to using an app to access cancer-related information for remote care during survivorship. Digital health literacy is linked to people’s ability to evaluate online health sources and their trust in the Internet as a health information source, according to a systematic review [25]. Our results show that low digital health literacy among cancer survivors could represent a major access barrier for the use and adoption of m-health tools that will be developed in the future [46]. The results of a binary logistic regression analysis showed that higher levels of self-perceived digital literacy were independent predictors of greater interest in obtaining health information during survivorship and more receptiveness of acceptability of using m-health apps to provide cancer supportive care. Studies have revealed that patients with higher digital/eHealth literacy are more likely to use an eHealth platform. Additionally, data on digital or eHealth literacy may provide more insight than sociodemographic data into the reasons why patients aren’t accessing digital health services [47]. A study by Lepore in 2019 [48] found that breast cancer survivors with low digital literacy may not benefit from digitally mediated care. The researchers also concluded that less digitally-literate women were less likely to participate in online support groups and had higher distress and anxiety levels. Therefore, if digital health apps fail to address the needs of marginalised groups, such as effective access to digital technology, they may make health inequities worse [49]. This is in line of our findings that cancer survivors use and uptake of health apps was the lowest, 14.7% compared to 38.9% and 99.6%, for services apps and social media platforms, respectively. Despite having smartphones and Internet access, patients with low digital literacy may not benefit as much from online resources or digital interventions for cancer supportive care. This extends the digital divide in cancer information, preventing patients from using digital solutions. These findings are supported by a previous study, which suggests that when designing innovative digital interventions for cancer survivors, digital literacy should be considered [48]. Our findings also confirm that older cancer patients had significantly lower digital literacy than younger patients, who are more likely to use and evaluate Internet information [25,50]. Furthermore, patients with less education have lower digital literacy. This is consistent with the literature, which shows that those with lower educational attainment have lower actual and self-rated digital literacy skills to evaluate Internet content, resulting in lower trust in online cancer information resources [23,25]. Several researchers expressed concerns about the potential consequences of using unregulated or low-quality online platforms [51,52]. Our findings show that digital illiteracy and information trustworthiness/reliability were the main barriers to online cancer information acquisition. Although the two concepts are closely related, reliability was defined as whether health-related content is supported by medical evidence and/or approved by medical professionals or experts and referred to regulated websites with known affiliations. Whether regulated or not, Internet information’s “trustworthiness” refers to its perceived credibility. This is consistent with previous studies, such as AlNaiem [7], who found that few Arabic-language websites provide reliable and trusted breast cancer information, and most of them provide poor or fair quality information. On the contrary, another study found that the quality of breast cancer information on English websites was generally good [53]. Investigating the information needs, typologies, and timings for cancer survivors was the goal of the first part of this project [3]. The findings demonstrated high information needs throughout survivorship, hence m-health apps were considered to fulfil the population’s information needs. Patients’ interest in cancer supportive information to manage their health and their receptivity to cancer information throughout survivorship and its connection to socioeconomic variables and digital literacy were investigated. Jordanian cancer survivors’ primary preferred source of information was their treating physician (48.1%) which corroborated with other studies on cancer patients [54,55]. In a Spanish study, cancer patients and carers reported a low use of the Internet for searching for medical information, despite the fact that it helps patients cope with cancer, as well as a reluctance of cancer patients and carers to disclose their search findings with their physicians. Thirty percent of patients felt more confused as a result of their Internet search [56]. Previous studies showed that obtaining cancer-related information via the Internet is unlikely to substitute a face-to-face visit with a health provider [54,57]. Only 22% of online users reported discussing information with clinicians during clinical encounters. Two studies found 24% and 39% and few studies found greater rates [56]. Fear was cited as the key factor, since patients can anticipate that their search will be interpreted as a challenge to the doctor’s authority. The patient–doctor relationship’s communication style may affect patient’s active participation by discussing information with clinicians. In Jordan, and other Arabic countries, medical paternalism shapes the patient-provider’s relationship. Although Jordanian doctors prioritise patient autonomy, self-determination, and the right to information, paternalism is valued as a source of information by cancer patients as well [58]. Additionally, Chua et al. [54] showed that information seeking pervades all stages of survivorship and information-seeking showed a gap between preferred and actual sources. The fact that people prefer health providers but use the internet and other sources suggests that this accessibility and interactivity issues need to be addressed. Concerns about information quality topped the list of information-seeking behaviours, so efforts to address this concern should not be spared. Thus, professionally-designed digital apps can provide tailored cultural and cognitive health education about medical treatment and improve patient–doctor communication [41]. Our study found that online information seeking decreased after survivorship, even though this population was highly interested in cancer information. Previous studies have shown that cancer patients’ information-seeking patterns may change over time [3,9]. The majority of patients developed their information needs early in the cancer continuum (e.g., immediately post diagnosis or within two months post diagnosis) [3]. Interestingly, more than two-thirds of patients (68.4%) indicated they are still interested in receiving self-management information via a dedicated mobile app. However, digital illiteracy, loss of motivation or interest in online resources, information inertia, the app being seen as a constant reminder of illness and preferring face-to-face communication with doctors were key reasons for not using dedicated mobile apps for cancer supportive care. Given the downsides of internet users’ behaviour when searching for cancer-related information, digital health literacy helps people use internet tools and information to improve their health. Digital health technology can help cancer survivors coordinate treatment and promote self-management by bridging the digital literacy gap. This is especially true given the limitations and inaccuracies of internet cancer information [59]. However, the deployment of digital health technologies like m-health and eHealth in oncology should address the heterogeneity of digital health literacy and user characteristics from different age groups to benefit cancer survivors along the care continuum [60]. Tailoring digital interventions like Patient Decision Aids (PDAs) and risk visualisation tools can help patients with low digital literacy actively participate in decision-making with their doctors [46]. Furthermore, educational interventions for cancer patients can be delivered through interactive media using digital interventions designed for varying levels of digital literacy and leveraging web-based delivery systems.Additionally, without careful strategic planning that accounts for patients’ engagement barriers, facilitators, requirements, and opportunities, digital health technologies may not only fail to improve health care equity in the context of cancer, but may actually exacerbate existing disparities [60]. As a result, all stakeholders, including vulnerable groups, health policymakers, healthcare providers and other stakeholders need to be involved in a meaningful way to establish an agenda to promote digital health literacy, devise ways to maximise digital inclusion, and develop digital health interventions that are more likely to be effective and serve the WHO’s goal of leaving no one behind [61]. Dynamic multilayer solutions are needed to accommodate different digital health literacy levels. On the other hand, due to the difficulty of boosting limited digital health literacy, mitigating approaches will likely centre on expanding staff to assist individuals who struggle [54]. Digital navigators and telehealth task forces are examples [62,63]. Digitally-skilled family members or informal carers with biomedical backgrounds and a high education can serve as mediators for patients with limited digital health literacy [46]

## 5. Conclusions

This study showed that breast and colorectal Jordanian cancer survivors are using online tools to fill information gaps. Patients’ assessments of online cancer information’s authenticity are unknown. Health care systems need to adapt for technology drivers by providing online health education materials that are evidence-based, controlled, reliable, and culturally and linguistically appropriate and bridging the digital divide for cancer patients with low digital literacy so they can use professionally created digital platforms. Digital health literacy should be integrated into all health communication and public health research in line with the WHO digital ecosystem. The frequency with which people access the Internet in search of health information may be indicative of their health literacy and their ability to take charge of their health. The COVID-19 pandemic exemplifies the vast amount of health information and suggestions that people are exposed to via various digital channels. As a result, people must be prepared with appropriate health literacy skills in order to identify the most relevant, trustworthy information and services and distinguish the correct from the erroneous. Developing solutions that do not make inequities worse or undermine equality in the digital age requires researchers to build digital patient platforms that are adaptable to varying levels of digital literacy.

## 6. Limitations

Despite that multiple measurements are available for assessing digital health literacy in Web-based health information contexts [64], this study measured digital health literacy using self-perception. Based on Norman and Skinner’s concept, the question assessed participants’ digital health literacy on a five-point Likert scale [29]. The study’s objectives were met by this aggregate question due to its large sample size, resource constraints, questionnaire length, and COVID-19 lockdown. The first author was properly trained and clarified the questions and answered participant inquiries, so a telephone-based survey was chosen. This was advantageous for cancer survivors who were elderly, illiterate, lacked health literacy, lived in rural or underserved areas, or were suffering from cognitive decline. Future studies should focus on conceptualising health literacy in Web contexts to improve operationalization and measurement.

## 7. Practice Implications

This study found elements that may influence online information seeking behaviour, digital trends, and online information access barriers. According to the survey, most cancer patients prefer digital cancer information, especially if it is regulated by experts. The findings also imply that inadequate digital literacy among cancer survivors in developing countries may hinder involvement with fast-growing digital interventions and make it difficult to use digital educational platforms. Therefore, unaddressed subpar digital literacy may increase the digital divide and inequities because a substantial percentage of patients may be disadvantaged. In the digital age, developing nations’ health care systems could improve patients’ digital literacy by incorporating family members in digital efforts. COVID-19 revolutionised survival care. Digital health solutions including patient health portal systems, e-consultations (telephone or video consultations), and telemedicine interventions are being rapidly adopted and deployed to address the post-pandemic’s tectonic change in healthcare delivery. Regardless of the possible benefits of technology in this situation, it is vital to remember that the COVID-19 crisis emphasises the need to digitalize healthcare services in the developing world, with a focus on improving the accessibility and usability of these technologies. The lack of digital literacy education in developing nations simply exacerbates existing disparities. The findings could help to shape the design of future digital interventions. The design of such interventions should be user-centric and accessible to those with little digital literacy and cyber experience. Digital equality and inclusion should also be promoted by providing end users with digital training and technical help to bridge the digital divide.

## Figures and Tables

**Figure 1 ijerph-20-01472-f001:**
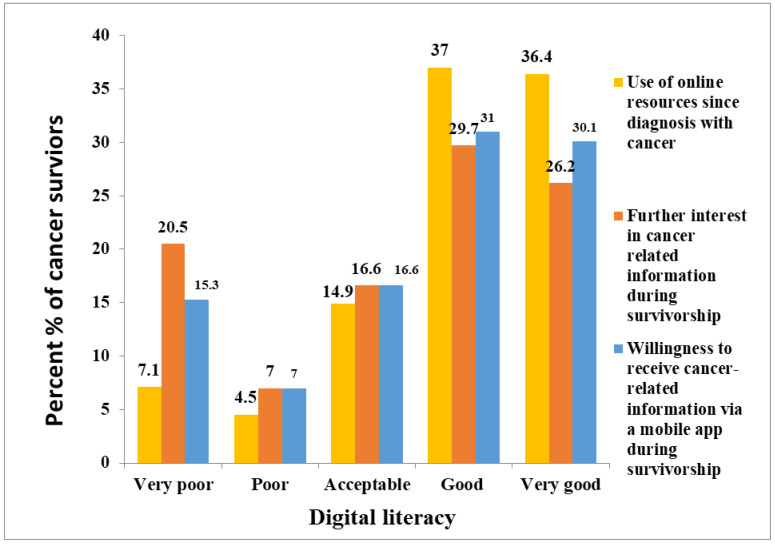
Digital literacy and digital trends of breast and colorectal cancer survivors (*n* = 335).

**Table 1 ijerph-20-01472-t001:** Proportionate sample size distribution per gender (male, female) and cancer primary distribution assuming a finite total population of 1567.

Cancer Type	Male (s)	Female (s)	Total
N	Eligible Sample (n)	ResponseSample(n)	N	Eligible Sample (n)	Response Sample(n)	N	Eligible Sample (n)	Response Sample(n)
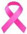 Breast	11	11	4	1142	292	251	1153	303	225
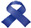 Colon	185	47	38	130	33	21	315	80	59
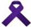 Rectum	52	14	12	47	12	9	99	26	21
Total	248	72	54	1319	337	281	1567 **	409 *	335 ◊

** Eligible population, * Total sample, ◊ Response sample.

**Table 2 ijerph-20-01472-t002:** Patient demographics stratified by online seeking behavior *n* = 335.

Socio-Demographic Characteristics (*n* = 335)	Online Resources Use for Cancer Related Information since Diagnosis		*p* Value
	Yes	No	
	*n* (%)	*n* (%)	*p*
**Gender**			0.05
Male	18 (33.3)	36 (66.7)	
Female	136 (48.4)	145(51.6)	
**Age**			0.00
less than 40	19 (86.4)	3 (13.6)	
40–49	45 (62.5)	27 (37.5)	
50–59	58 (50.0)	58 (50.0)	
60–69	23 (33.8)	45 (66.2)	
70+	9 (15.8)	48 (84.2)	
**Regions of residence**			0.10
Middle region	127 (48.1)	137 (51.9)	
North region	20 (33.9)	39 (66.1)	
South region	7 (58.3)	5 (41.7)	
**Employment (paid or unpaid)**			0.00
Housewife	43 (75.4)	14 (24.6)	
Retired	83 (40.7)	121 (59.3)	
Student	0 (0.0)	0 (0.0)	
Employed (paid or unpaid)	28 (41.2)	40 (58.8)	
Employed (paid or unpaid)	0 (0.0)	6 (100.0)	
**Monthly income in Jordanian Dinars (US $)**			0.00
less than 100 JOD (< 140$)	2 (12.5)	14 (87.5)	
100–199 JOD (140-280$)	9 (37.5)	15 (62.5)	
200–299 JOD (282-422$)	15 (35.7)	27 (64.3)	
300–499 JOD (423-704$)	23 (36.5)	40 (63.5)	
500 JOD or more (≥705$)	44 (71.0)	18 (29.0)	
I don’t know	35 (45.5)	42 (54.5)	
Refuse to answer	26 (51.0)	25 (49.0)	
**Education highest level of Education**			0.00
Illiterate	0 (0.0)	26 (100.0)	
Elementary school	11 (14.1)	67 (85.9)	
High school	47 (51.6)	44 (48.4)	
Diploma	38 (56.7)	29 (43.3)	
Bachelor’s degree	49 (77.8)	14 (22.2)	
Masters/PhD	9 (90.0)	1 (10.0)	
I don’t know	0	0	
Refuse to answer	0	0	
**Type of cancer**			0.02
Breast	128 (50.2)	127 (49.8)	
Colon	20 (33.9)	39 (66.1)	
Rectum	6 (28.6)	15 (71.4)	
**Chronic Disease (diabetes, hypertension, cardiovascular disease, Others)**			0.00
Yes	71 (29.8)	168 (70.2)	
No	98 (30.0)	77 (70.0)	

JOD: Jordanian Dinar.

**Table 3 ijerph-20-01472-t003:** Distribution of the respondents according to action taken to check reliability and trustworthiness of the cancer health information acquired from the Internet.

Action Taken to Verify Cancer Health Information	*n* (%)
Asking doctor or a health professional	97 (63.0)
Verify results on other websites	39 (25.3)
Check other information sources	10 (6.5)
Ask the opinion of others	15 (9.7)
Do nothing	28 (18.2)

**Table 4 ijerph-20-01472-t004:** Association between demographic characteristics and digital literacy of breast and colorectal survivors.

Variables	Digital Health Literacy					
	Very Poor	Poor	Acceptable	Good	Very Good	*p*-Value
**Sex**						0.884
Male	19 (35.20)	4 (7.40)	7 (13.00)	11 (20.40)	13 (24.10)	
Female	91 (32.40)	18 (6.40)	39 (13.90)	75 (26.70)	58 (20.60)	
**Age group(years)**						0.000
≤40	1 (9 %)	0 (0.0)	3 (6.5)	11 (12.8)	7 (9.9)	
40–49	13 (11.8)	6 (27.3)	7 (15.2)	21 (24.4)	25 (35.2)	
50–59	23 (20.9)	12 (54.5)	21 (45.7)	31 (36.0)	29 (40.8)	
60–69	30 (27.3)	2 (9.1)	12 (26.1)	16 (18.6)	8 (11.3)	
≥70	43 (39.1)	2 (9.1)	3 (6.5)	7 (8.1)	2 (2.8)	
**Cancer type**						0.546
Breast	78 (30.60)	18 (7.1)	37 (14.5)	67 (26.3)	55 (21.6)	
Colon	25 (42.40)	4 (6.8)	5 (8.5)	12 (20.3)	13 (22.0)	
Rectum	7 (33.30)	0 (0.0)	4 (19.0)	7 (33.3)	3 (14.3)	
**Residence**						0.58
Middle region	85 (32.2)	15 (5.7)	38 (14.4)	67 (25.4)	59 (22.3)	
North region	23 (39.0)	6 (10.2)	7 (11.9)	15 (25.4)	8 (13.6)	
South region	2 (16.7)	1 (8.3)	1 (8.3)	4 (33.3)	4 (33.3)	
**Marital status**						0.032
Single	4 (21.1)	1 (5.3)	2 (10.5)	5 (26.3)	7 (26.8)	
Married	83 (30.2)	17 (6.2)	43 (15.6)	71 (25.8)	61 (22.2)	
Divorced	2 (66.7)	0 (0.0)	0 (0.0)	0 (0.0)	1 (33.3)	
separated	3 (33.3)	1 (11.1)	0 (0.0)	5 (55.6)	0 (0.0)	
Widowed	18 (62.1)	3 (10.3)	1 (3.4)	5 (17.2)	2 (6.9)	
**Employment status**						0.000
Employed (paid or unpaid	4 (7.0)	2 (3.5)	10 (17.5)	17 (29.8)	24 (42.1)	
Unemployed (capable or in capable)	5 (83.3)	1 (16.7)	0 (0.0)	0 (0.0)	0 (0.0)	
Housewife	79 (38.7)	16 (7.8)	26 (12.7)	53 (26.0)	30 (14.7)	
Student	0 (0.0))	0 (0.0)	0 (0.0)	0 (0.0)	0 (0.0)	
Retired	22 (32.4)	3 (4.4)	10 (14.7)	16 (23.5)	17 (25.0)	
Refuse to answer	0 (0.0)	0 (0.0)	0 (0.0)	0 (0.0)	0 (0.0)	
**Monthly income in Jordanian Dinars (US$)**						0.000
Less than 100 (140)	11 (68.8)	1(6.3)	0 (0.0)	3 (18.8)	1 (6.3)	
100–199 (140–280)	12 (50.0)	2 (8.3)	4 (16.7)	3 (12.5)	3 (12.5)	
200–299 (281–421)	17 (40.5)	4 (9.5)	5 (11.9)	6 (14.3)	10 (23.8)	
300–499 (422–703)	20 (31.7)	1 (1.6)	16 (25.4)	17 (27.0)	9 (14.3)	
500 (704) or more	9 (14.5)	2 (3.2)	8 (12.9)	17 (27.4)	26 (41.9)	
Don’t know	24 (31.2)	6 (7.8)	9 (11.7)	25 (32.6)	13 (16.9)	
Refuse to answer	17 (33.3)	6 (11.8)	4 (7.8)	15 (29.4)	9 (17.6)	
**Education status**						0.000
Illiterate	24 (92.3)	1 (3.8)	1 (3.8)	0 (0.0)	0 (0.0)	
Elementary school	44 (58.4)	9 (11.5)	11 (14.1)	7 (9.0)	7 (9.0)	
High school (Tawjihi)	21 (23.1)	8 (8.8)	18 (19.8)	32 (25.2)	12 (13.2)	
Diploma	13 (19.4)	2 (3.0)	10 (14.9)	24 (35.8)	18 (26.9)	
University /bachelor’s degree	7 (11.1)	2 (3.2)	6 (9.5)	20 (31.7)	28 (44.4)	
Masters/PhD	1 (10.0)	0 (0.0)	0 (0.0)	3 (30.0)	6 (60.0)	

**Table 5 ijerph-20-01472-t005:** Digital trends of breast cancer survivors and digital literacy (*n* = 335).

Variables	Use of Online Resources since Diagnosis with Cancer *n* = (%)		*p*	Interest to Receive Cancer-Related Information during Survivorship *n* = (%)		*p*	Willingness to Use a Mobile a Dedicated App to Receive Cancer-Related Information during Survivorship *n* = (%)		*p*-Value
Digital health literacy	Yes (*n* = 154) %	No (*n* = 181) %	0.00	Yes (*n* = 229) %	No (*n* = 106) %	0.00	Yes (*n* = 229) %	No (*n* = 106) %	0.00
Very poor	11 (7.1)	99 (54.7)		47 (20.5)	63 (59.4)		35 (15.3)	75 (70.8)	
Poor	7 (4.5)	15 (8.3)		16 (7.0)	6 (5.7)		16 (7.0)	6 (5.7)	
Acceptable	23 (14.9)	23 (12.7)		38 (16.6)	8 (7.5)		38(16.6)	8 (7.5)	
Good	57 (37.0)	29 (16.0)		68 (29.7)	18 (17.0)		71 (31.0)	15(14.2)	
Very good	56 (36.4)	15 (8.3)		60 (26.2)	11 (10.4)		69 (30.1)	2 (1.9)	

**Table 6 ijerph-20-01472-t006:** Predictors of online information interest across the cancer continuum among breast and colorectal cancer survivors, Jordan 2020, * *p*-value < 0.05 indicates statistical significance.

Variable (s)		Interest in Receiving Health Information during Survivorship		COR (95% CI)		AOR (95% CI)	*p* Value
		YES, *n* = (229)	No; *n* = (106)				
**Gender**							
Male		33 (9.9%)	21 (6.3%)	1.571 (0.909, 2.716)		1.295 (0.429, 3.908)	**0.646**
Female		196 (58.5%)	85 (25.4%)	*Referent*			
**Age (years)**							**0.395**
<40		18 (5.4%)	4 (1.2%)	4.500 (1.523, 13.296)		0.997 (0.144, 6.882)	0.997
40-49		58 (17.3%)	14 (4.2%)	4.143 (2.311, 7.426)		1.362 (0.409, 4.541)	0.615
50-59		82 (24.5%)	34 (10.1%)	2.412 (1.1617, 3.597)		0.555 (0.189, 1.632)	0.285
60-69		48 (14.3%)	20 (6.0%)	2.400 (1.425, 4.043)		1.303 (0.432, 3.927)	0.638
≥70		23 (6.9%)	34 (10.1%)	*Referent*			
**Education**							**0.395**
Illiterate		7 (2.1%)	19 (5.7%)	0.368 (0.155, 0.876)		0.288 (0.33, 2.513)	0.260
Elementary School		42 (12.5%)	36 (10.7%)	1.167 (0.748, 1.821)		1.212 (0.210, 6.984)	0.829
High School		69 (20.6%)	22 (6.6%)	3.136 (1.941, 5.068)		1.519 (0.279, 8.280)	0.629
Diploma		50 (14.9%)	17 (5.1%)	2.941 (1.696, 5.099)		1.697 (0.299, 9.644)	0.551
University Degree		53 (15.8%)	10 (3.0%)	5.300 (2.697, 10.417)		3.575 (0.583, 21.936)	0.169
Masters/PhD		8 (2.4%)	2 (0.6%)	*Referent*			
**Type of cancer**							**0.337**
Breast		183 (54.6%)	72 (21.5%)	2.542 (1.935, 3.338)		1.608 (0.428, 6.043)	0.482
Colon		33 (9.9%)	26 (7.8%)	1.269 (0.759, 2.122)		0.723 (0.202, 2.585)	0.618
Rectum		13 (3.9%)	8 (2.4%)	*Referent*			
**Monthly Income**							**0.768**
<100 JOD (<140 $)		9 (4.3%)	7 (3.4%)	1.286 (0.479, 3.452)		1.245 (0.25, 6.208)	0.790
100-199 JOD (140–279 $)		14 (6.8%)	10 (4.8%)	1.400 (0.622, 3.152)		0.573 (0.144, 2.272)	0.428
200-299 JOD (280–419 $)		26 (12.6%)	16 (7.7%)	1.625 (0.872, 3.029)		0.610 (0.194, 1.919)	0.398
300-499 JOD (749–500$)		44 (21.3%)	19 (9.2%)	2.316 (1.352, 3.966)		0.686 (0.232, 2.033)	0.497
≥500 JOD (750$)		51 (24.6%)	11 (5.3%)	*Referent*			
Region							**0.228**
North Region		185 (55.2%)	79 (23.6%)	2.342 (1.799, 3.048)		3.697 (0.821, 16.655)	0.089
Middle Region		36 (10.7%)	23 (6.9%)	1.565 (0.928, 2.641)		2.990 (0.564, 15.850)	0.198
South Region		8 (2.4%)	4 (1.2%)	*Referent*			
**Digital health literacy**							**0.006** *
Very poor		47 (14.0%)	63 (18.8%)	0.746 (0.511, 1.088)		0.194 (0.060,0.623)	0.006 *
Poor		16 (4.8%)	6 (1.8%)	2.667 (1.043, 6.815)		0.730 (0.127, 4.199)	0.725
Acceptable		38 (11.3%)	8 (2.4%)	4.750 (2.216, 10.181)		1.362 (0.338, 5.493)	0.664
Good		68 (20.3%)	18 (5.4%)	3.778 (2.247, 6.351)		0.828 (0.253, 2.709)	0.755
Very good		60 (17.9%)	11 (3.3%)	*Referent*			
**Omnibus Tests of Model Coefficients**							
			**Chi-square**	**df**	**Sig.**		
Step 1	Step		88.031	22	0.000		
	Block		88.031	22	0.000		
	Model		88.031	22	0.000		
	**Model Summary**						
Step 1	−2 Log likelihood		**Cox & Snell R Square**	**Nagelkerke R Square**			
	198.932a		0.346	0.462			

JOD: Jordanian Dinar, *p* value < 0.05 indicates significance and designated as (*).

**Table 7 ijerph-20-01472-t007:** Predictors of m-health app adoption for self-management in breast and colorectal cancer survivors, Jordan 2020, * *p*-value <0.05 indicates statistical significance.

Variable (s)	Willingness to Use M-Health App/Portal for Self-Management		COR (95% CI)	AOR (95% CI)	*p* Value
	YES, *n* = (229)	NO; *n* = (106)			
**Gender**					
Male	35 (10.4%)	19 (5.7%)	1.842 (1.054, 3.220)	1.469 (0.413, 5.233)	0.553
Female	194(57.9%)	87(26.0%)	*Referent*		
**Age**					**0.218**
<40	20 (6.0%)	2 (0.6%)	10.00 (2.337, 42.783)	5.985 (0.460, 77.920)	0.172
40–49	55 (16.4%)	17 (5.1%)	3.235 (1.878, 5.573)	2.032 (0.566, 7.295)	0.277
50–59	90 (26.9%)	26 (7.8%)	3.462 (2.237,5.355)	3.386 (1.030, 11.130)	0.045
60–69	46 (13.7%)	22 (6.6%)	2.091 (1.258,3.475)	3.353 (1.039, 10.824)	0.043
≥70	18 (5.4%)	39 (11.6%)	*Referent*		
**Education**					**0.783**
Illiterate	5 (1.5%)	21 (6.3%)	0.238 (0.090, 0.631)	0.278 (0.022, 3.476)	0.321
Elementary School	39 (11.6%)	39 (11.6%)	1.000 (0.642, 1.559)	0.567 (0.065, 4.943)	0.607
High School	71 (21.2%)	20 (6.0%)	3.550 (2.161, 5.831)	0.828 (0.102, 6.715)	0.859
Diploma	51 (15.2%)	16 (4.8%)	3.187 (1.818, 5.589)	0.605 (0.72, 5.078)	0.644
University Degree	53 (15.8%)	10 (3.0%)	5.300 (2.697, 10.417)	1.113 (0.127, 9.767)	0.923
Masters/PhD	10 (3.0%)	0 (0.0%)	*Referent*		
**Type of cancer**					**0.504**
Breast	179 (53.4%)	76 (22.7%)	2.355 (1.801, 3.080)	2.174 (0.517, 9.142)	0.289
Colon	37 (11.0%)	22 (6.6%)	1.682 (0.992, 2.851)	2.216 (0.521, 8.679)	0.293
Rectum	13 (3.9%)	8 (2.4%)	*Referent*		
**Monthly Income**					**0.874**
<100 JOD (<140$)	9 (4.3%)	7 (3.4%)	1.286 (0.479, 3.452)	2.061 (0.362, 11.718)	0.415
100–199 JOD (140–279$)	14 (6.8%)	10 (4.8%)	1.400 (.622, 3.152)	1.319 (0.273, 6.373)	0.730
200–299 JOD (280–419$)	26 (12.6%)	16 (7.7%)	1.625 (0.872, 3.029)	0.897 (0.251, 3.208)	0.867
300–499 JOD (749–500$)	41 (19.8%)	22 (10.6%)	1.864 (1.110, 3.128)	0.989 (0.312, 3.133)	0.985
≥500 JOD (750$)	52 (25.1%)	10 (4.8%)	*Referent*		
Region					**0.029** *
**Middle Region**	186 (55.5%)	78 (23.3%)	2.385 (1.831, 3.106)	13.285 (1.793, 98.414)	0.011 *
North Region	34 (10.1%)	25 (7.5%)	1.360 (0.811, 2.279)	7.382 (0.907, 60.096)	0.062
South Region	9 (2.7%)	3 (0.9%)	*Referent*		
**Digital health literacy**					**0.000 ***
Very poor	35 (10.4%)	75 (22.4%)	0.467 (0.312, 0.697)	0.013 (0.002, 0.112)	0.000 *
Poor	16 (4.8%)	6 (1.8%)	2.667 (1.043, 6.815)	0.052 (0.004, 0.635)	0.021 *
Acceptable	38 (11.3%)	8 (2.4%)	4.750 (2.216, 10.181)	0.089 (0.101, 0.816)	0.032 *
Good	71 (21.2%)	15 (4.5%)	4.733 (2.712, 8.261)	0.082 (0.010, 0.658)	0.019 *
Very good	69 (2.06%)	2 (0.6%)	*Referent*		
**Omnibus Tests of Model Coefficients**					
		**Chi-square**	**df**	**Sig.**	
Step 1	Step	116.632	22	0.000	
	Block	116.632	22	0.000	
	Model	116.632	22	0.000	
**Model Summary**					
Step 1	−2 Log likelihood	**Cox & Snell R Square**	**Nagelkerke R Square**		
	170.331a	0.431	0.574		

JOD: Jordanian Dinar, *p* value < 0.05 indicates significance and designated as (*).

## Data Availability

Data and resources are available upon request. Coding data are available upon request from the corresponding author.

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
