# Peer review of "Digital Trends, Digital Literacy, and E-Health Engagement Predictors of Breast and Colorectal Cancer Survivors: A Population-Based Cross-Sectional Survey"

_ijerph, 2023, doi:10.3390/ijerph20021472_

Round 1

Reviewer 1 Report

Authors present a question regarding the application of digital literacy into online health-related research.  As a manifestation of the "WebMD effect" where online health research done without appropriate training or context often leads to negative patient-physician interactions and eventual health outcomes, this is a salient question and one with value beyond academia into the practical.  The COVID-19 pandemic and the shift into more online modes of consumption and communication may have exacerbated such a shift.  Into that context the authors inquire whether a high-demand population of breast, colon, and rectal cancer survivors, exhibit different patterns of consumption of online health information based on demographic characteristics and digital literacy.

Functionally, the authors need to provide some sense of what is novel about these findings.  For the most part, the conclusion is that lower levels of digital literacy lead to less quantity and quality of online information search, which is intuitive.  All of the the conclusions drawn are consistent with that thesis, thus the authors need to show what the substantive contribution of new knowledge is.  

Since the target population is cancer survivors, there are naturally some biases within any sample that are difficult to adjust for.  Particularly, the preponderance of breast cancer patients making a fivefold discrepancy over colon and rectal, makes the non-breast-cancer portion of the sample suspect.  I would suggest significantly increasing (if possible) the proportion of the sample including colon and rectal cancer patients, and if that is not possible reframing the survey to focus only on breast cancer would be appropriate.  The large proportion with breast cancer brings concomitant biases towards female and older patients as well.  In other words, there is a cascade of intrinsic biases introduced by such a highly-skewed sample of breast cancer patients.  

Turning to the specific findings, a few areas need attention.  On line 263, the authors note that reliability and trustworthiness of online information searches were significant obstacles for respondents.  Since those two topics are closely related, it is highly possible these two variables are correlated.  I suggest running another inferential statistical test to determine if there is an interaction effect between the two perceptions and reduced impetus to search for health-related information online.  

Using the self-perception of digital literacy is an imperfect measure and could be improved.  A more substantive measure, particularly among older respondents, would be more reliable as a metric for digital literacy.  The Mobile Device Proficiency Questionnaire (MDPQ, see  Roque, N. A., & Boot, W. R. (2018). A New Tool for Assessing Mobile Device Proficiency in Older Adults: The Mobile Device Proficiency Questionnaire. Journal of Applied Gerontology37(2), 131–156. https://doi.org/10.1177/0733464816642582
for more details) is one system that would give more confidence in the level of one's individual digital literacy.  

As submitted, the paper provides some interesting findings.  On lines 321 and 322, authors find that half of patients preferred obtaining cancer-related information directly from doctors.  This inspires two questions/comments: 1) this would seem to drive why the m-health app intervention was used, but the authors should clarify that connection in the narrative, and 2) the finding makes the Jordanian sample stand out as different from other nations such as the United States of America (see  https://www.fiercehealthcare.com/practices/americans-don-t-understand-provider-information-seek-help-from-internet) and deserves some more investigation as to what factors may contribute to why Jordan deviates from the norm here. 

On lines 349 and 350, the authors report that digital literacy was the only independent predictor of cancer-related information acquisition during survivorship.  Considering the demographic differences in digital literacy reported earlier, this is a significant finding.  This could mean that older populations have less access to information and thus are less prepared to discuss health care options with their physicians, or it could mean that younger and more digitally literate patients are being misinformed by the less reliable and trustworthy information they find online.  Authors should discuss the ramifications of this and identify which one of those two theories is more supported by the data.  

Some points of clarification would be helpful here.  In lines 138-140, authors note four sections of questions subdivided by topic: "online information access and search, cancer supportive care information, and survivorship information openness."  That reads as three topic areas.  If information access and search are two separate topical sections, they should be broken out as such. If instead there is a fourth category not listed, it should be added.  

One significant opportunity for this manuscript is a deeper discussion of the implications of the digital literacy gap on sought standards of care.  Do patients who seek more information online prepare themselves better to discuss options with their physician, or as in the link about US patients they find themselves more confused and conflicted between what they consume online and what they discuss with their physicians.  The digital literacy gap in online health information seeking can influence standards of patient care for cancer survivors, and as age is highly correlated with that gap the implications for skewed health care outcomes by age are significant.  

Author Response

Dear Reviewer

Kind regards

Reviewer 2 Report

The authors presented at an approirate style and ways. The study is original and relevant to digital access to cancer care.

1. It is suggested to incorporate the survey as supplementary materials to give audience a better perception of the survey design. It would be a valuable reference for future researchers too.
2. The authots are suggested to unify the table format (table 4 vs others).
3. There're duplicated sections of acknowledge and conflict of interest. please remove either one.

Author Response

Dear Reviewer please see the attahment

Kind regards

Round 2

Reviewer 1 Report

My compliments to the authors for a comprehensive response to my review and for the substantive improvements made.  The new revision allays my concerns and I believe the manuscript is ready to publish now.